Association of chromosomal aberrations in chromosomes 3 and 7, and P16 mutations with malignancy in salivary gland tumors

Wu Yanxia 1 2 3
Xie Zhiyue 1 2
http://orcid.org/0009-0004-2284-2265 Peng Nan 2
Zhou Rui 2
Zhao Liang 1 2 liangsmu@foxmail.com
1 Department of Pathology, Nanfang Hospital, Southern Medical University , Guangzhou, Guangdong , China
2 Department of Pathology, Guangdong Province Key Laboratory of Molecular Tumor Pathology, School of Basic Medical Sciences, Southern Medical University , Guangzhou, Guangdong , China
3 Department of Pathological Diagnosis and Research Center, Affiliated Hospital of Guangdong Medical University , Guangzhou, Guangdong , China
Ateş Mehmet Burak
Electronic publication date: 2025 Mar 31
Publication date: 2025
Volume: 13
Electronic Location ID: e19217
Received 2024 Oct 11; Accepted 2025 Mar 5
Copyright: © 2025 Wu et al.
Copyright year: 2025
Copyright holder: Wu et al.
License: This is an open access article distributed under the terms of the Creative Commons Attribution License, which permits unrestricted use, distribution, reproduction and adaptation in any medium and for any purpose provided that it is properly attributed. For attribution, the original author(s), title, publication source (PeerJ) and either DOI or URL of the article must be cited.
License URL: https://creativecommons.org/licenses/by/4.0/

Keywords: Salivary gland tumors, Pathological diagnosis, Fluorescence in situ hybridization

Funding: National Natural Science Foundation of China 82073026 This work was supported by the National Natural Science Foundation of China (82073026). The funders had no role in study design, data collection and analysis, decision to publish, or preparation of the manuscript.

==============================
Background

Salivary gland tumors, a prevalent type of head and neck neoplasm, exhibit significant morphological diversity and overlapping features, complicating pathological diagnosis. Although fluorescence in situ hybridization (FISH) is widely used for tumor detection, its diagnostic utility in salivary gland tumors remains unclear. This study aimed to explore a novel FISH-based approach to differentiate benign from malignant salivary gland tumors.

Methods

Gene probes (CSP3, CSP7, and GSP P16) were designed to detect P16 gene deletion, and polysomies of chromosomes 3 and 7. The FISH analysis was conducted on 22 malignant and 12 benign salivary gland tumor samples with complete clinical data. The study was expanded to 78 samples for further validation.

Results

The CSP3, CSP7, and GSP P16 probes exhibited high specificity for salivary gland tumors, though CSP7 exhibited lower sensitivity. The combination of CSP3 and GSP P16 probes outperformed single-probe analysis or other probe combinations.

Conclusion

The CSP3 and GSP P16 probe combination provides a highly sensitive and specific method for distinguishing malignant from benign salivary gland tumors.

Introduction

Salivary gland tumors are rare neoplasms arising from salivary gland tissues, accounting for less than 5% of head and neck tumors. These tumors exhibit significant heterogeneity and complex clinical presentations (Rached et al., 2024; Weaver et al., 2023). Despite a notable rise in incidence, their etiology remains poorly understood, and there are no systematic preventive strategies or standardized treatment protocols. Managing recurrent or metastatic cases remains controversial (Lin, Limesand & Ann, 2018; Wang et al., 2017; Geiger et al., 2021). Diagnostic challenges persist, even with the widespread use of fine-needle aspiration biopsy (Wakely, 2022; Maleki et al., 2018). The complex developmental and morphological characteristics of normal salivary glands contribute to diverse cellular morphologies, tissue structures, and biological behaviors of salivary gland tumors. According to the WHO Classification of Head and Neck Tumors (5th edition), there are over 20 malignant and 15 benign primary salivary gland tumor types (Mete & Wenig, 2022; Skálová, Hyrcza & Leivo, 2022). Some low-grade malignancies initially exhibit benign features and progress slowly, while variability in morphological grading among pathologists further complicates the differentiation between benign and malignant tumors (Weaver et al., 2023; Jalaly & Baloch, 2023). These factors significantly impact treatment efficacy and long-term patient outcomes.

Fluorescence in situ hybridization (FISH) enables direct visualization of specific genes and their precise spatial localization within cells, making it a valuable tool for detecting chromosomal and genetic abnormalities (Almeida & Azevedo, 2021; Mahendru et al., 2023). Owing to its high sensitivity, specificity, and simplicity, FISH is extensively employed in diagnosing malignancies, such as urothelial carcinoma, hematologic malignancies, breast cancer, and cervical cancer (Quessada et al., 2021; Zito Marino et al., 2021; Chen et al., 2020; Rigon et al., 2015; Rotman et al., 2020). Recent studies have identified chromosomal abnormalities as potential risk factors for salivary gland tumors. Polysomy of chromosome three has been observed in adenoid cystic carcinoma (Götte et al., 2005) and pleomorphic adenomas (Li et al., 1995), while chromosome seven polysomy has been reported in adenocarcinomas (Götte et al., 2005) and salivary duct carcinoma (Williams et al., 2010). However, limited research has explored chromosomal differences between benign and malignant salivary gland tumors. Emerging evidence suggests that NOTCH1 (chromosome 3) and EGFR (chromosome 7) play key roles in the development and progression of malignant salivary gland tumors (Ho et al., 2019; Zhou et al., 2023; Guazzo et al., 2021). Investigating polysomy on chromosomes 3 and 7 in salivary gland tumors warrants further exploration.

Chromosomal aneuploidy is a common genetic alteration in tumors, with polysomies of chromosomes 3 and 7 frequently observed. These alterations have been extensively studied in cancers such as urothelial carcinoma, esophageal squamous cell carcinoma, and cervical cancer (Wang et al., 2013; Li et al., 2011; Chen et al., 2017). The P16, located on chromosome 9p21, is a key regulator of the cell cycle, controlling cell proliferation and division. Mutations in P16 are early events in various cancers, including mesothelioma, breast cancer, oropharyngeal squamous cell carcinoma, and ovarian cancer (Lebok et al., 2016; Ito et al., 2015; Dacic, 2022; Berdugo, Rooper & Chiosea, 2021). The detection of chromosome 3 and 7 abnormalities, and P16 deletion using FISH technology have been widely applied in diagnosing and monitoring bladder cancer (Diao et al., 2020; Zheng et al., 2023). However, its role in diagnosing and differentiating salivary gland tumors remains unexplored. This study investigated the utility of FISH analysis in distinguishing benign from malignant salivary gland tumors, providing clinically relevant insights for pathologists.

Materials and Methods

Clinical samples

Surgically excised salivary gland tumor and benign/normal salivary gland tissue specimens were collected from the affiliated hospital of Guangdong Medical University. Based on the 5th edition of the WHO classification for salivary gland tumors, the samples were categorized into 38 malignant and 40 benign salivary gland tumors or normal salivary gland tissues. The malignant salivary gland tumor samples included 20 cases of mucoepidermoid carcinoma, 13 of adenoid cystic carcinoma, two of salivary duct carcinoma, two of myoepithelial carcinoma, and one of secretory carcinoma, which is not classified in the 5th edition of the WHO classification. The benign salivary gland tumor samples comprised 19 cases of pleomorphic adenoma, 11 of Warthin’s tumor, one of basal cell adenoma, and nine of benign/normal salivary gland tissue. Tumors of non-salivary gland origin were excluded. Tissue sections were cut to a thickness of 5 μm. Peripheral blood cells were collected from healthy volunteers with no history or presence of cardiovascular, renal, hepatic, pulmonary, hematological, gastrointestinal, metabolic, endocrine, immunological, neurological, or psychiatric diseases. All procedures adhered to the Declaration of Helsinki and were approved by the Ethics Committee of Guangdong Medical University (number PJKT2024-196).

Construction of FISH probes

The CSP3 and CSP7 probes were designed based on the high-specificity centromeric regions of human chromosomes 3 and 7. Primer sequences for the probes are listed in Table S1. The P16 gene sequence was selected from Invitrogen’s RP11 BAC clone library (Table S2), with plasmids extracted using a commercial kit (Takara Biomedical Technology, Dalian, China). Primers were synthesized by Sangon Biotech Co. Ltd. (Shanghai, China), with human genomic and plasmid DNA as the templates for polymerase chain reaction (PCR) amplification. The PCR products were analyzed using 2% agarose gel electrophoresis. The gap-filling method was adopted for fluorescent labeling of the amplified products, with tetramethylrhodamine (TAMRA, Roche Diagnostics, Indianapolis, IN, USA) as the fluorophore. The reaction mixture was prepared, mixed, centrifuged on ice under strict light protection, and labeled at 25 °C for 2 h. Enzyme inactivation occurred at 80 °C for 10 min. The labeled products were precipitated with sodium acetate and purified by high-speed centrifugation.

FISH analysis

Slides were preheated at 65 °C for 5–30 min. Deparaffinization was performed at room temperature using an environmentally friendly agent (Guangzhou Anbiping Pharmaceutical Technology Co., Ltd., Yuexiu, China) for 10 min. Slides were immersed in room-temperature anhydrous ethanol to remove residual agents for 10 min. Rehydration was performed sequentially in anhydrous ethanol, 90% ethanol, 70% ethanol, and purified water, each for 3 min. Samples were boiled in purified water at 95–100 °C for 25 min, air-dried, and digested with 100–200 μL pepsin working solution at 37 ± 1 °C for 3–15 min. After washing in 2 × SSC (saline-sodium citrate) at room temperature for 3 min, slides were dehydrated sequentially in 70%, 90%, and 100% ethanol at room temperature for 2 min each, then air-dried. Approximately 4–8 μL hybridization solution was applied to the hybridization area, covered with a coverslip, and gently pressed to ensure uniform probe distribution and bubble removal. The coverslip was sealed with rubber cement, denatured at 85 ± 1 °C for 5 min, and hybridized at 37 ± 1 °C for 10–18 h. After gently removing the rubber cement and coverslip, slides were washed at 2 × SSC at 37 ± 1 °C for 10 min, followed by washing in 0.1% nonylphenol (NP)-40/2 × SSC at 37 ± 1 °C for 5 min. The slides were immersed in 70% ethanol at room temperature for 3 min, then stained with 5–7 μL of 4′,6-diamidino-2-phenylindole (DAPI) after air-drying. The slides were examined under a fluorescence microscope after sealing with a coverslip in the dark.

Statistical evaluation

Fluorescent signals of DAPI and probes were observed, with CSP7 and GSP P16 probes emitting red fluorescence, while CSP3 probes emitted green fluorescence. Fifty cells with distinct cell contours, uniform DAPI staining, clear probe signals, and no overlapping nuclei were selected. The nucleus’s green and red fluorescent signals were counted at 1,000x magnification, and the ratio of abnormal to normal cells was calculated. Each sample was evaluated by counting the same number of cells, and the percentage of abnormal signal points was recorded. A sample was considered positive if the rate of abnormal signal points exceeded the threshold (10% for chromosome 3/7 abnormalities and 20% for P16 gene abnormalities); otherwise, it was deemed negative. If the percentage equaled the threshold, additional cell observations were required. Sensitivity was defined as the concordance rate between the probe-diagnosed positive samples and pathological diagnosis positive samples (malignant salivary gland tumors). Conversely, specificity was defined as the concordance rate between probe-diagnosed negative samples and pathological diagnosis negative samples (normal/benign salivary gland tissue).

Results

Validation of CSP3, CSP7, and GSP P16 probes

To assess the sensitivity and specificity of the designed probes, they were tested on human peripheral blood cultured cells. Fifty metaphase cells were analyzed to evaluate the fluorescence intensity, hybridization efficiency, and accuracy of hybridization sites. The gene loci corresponding to CSP3 and GSP P16 probes were labeled with green and red fluorescence, respectively. The fluorescent signals were clear, with no cross-hybridization, indicating a 100% (50/50) hybridization efficiency (Figs. 1A–1D).

Figure 1 Validation of FISH probes in cultured human peripheral blood cells.

Representative images of FISH using CSP3 (green, A), GSP P16 (red, B), the merged image of CSP3 and GSP P16 (C), and CSP7 (red, D) probes in metaphase of cultured peripheral blood cells.

Application of CSP3, CSP7, and GSP P16 probes in salivary gland tumors

FISH probes were applied to 34 salivary gland tumor samples with confirmed pathological diagnoses, including 22 malignant tumors (mucoepidermoid carcinoma and adenoid cystic carcinoma) and 12 benign tumors. The results were as follows: CSP3 showed 17 true positives, five false positives, 10 true negatives, and two false negatives; CSP7 revealed seven true positives, 15 false positives, 11 true negatives, and one false negative; GSP P16 demonstrated 17 true positives, five false positives, 10 true negatives, and two false negatives (Table S3). The FISH results were analyzed (Table 1), with representative images displayed (Figs. 2A–2B & 3A–3D). Moreover, the sensitivity and specificity of the CSP3, CSP7, and GSP P16 probes, and their combinations in distinguishing benign from malignant tumors, were calculated (Table 2). The results indicated that CSP3 probes exhibited high specificity (>80%) for salivary gland tissues, whereas CSP7 probes had relatively low sensitivity (31.8%). Moreover, the CSP3 and GSP P16 combination yielded the best performance, with 90.9% sensitivity and 83.3% specificity, outperforming individual probes and other combinations, thereby offering superior detection for malignant salivary gland tumors.

Table 1 Evaluation of FISH with CSP3, CSP7, and GSP P16 probes in 34 salivary gland tumor tissue samples (22 malignant and 12 benign tumors).

Pathological diagnosis	Count	FISH results	
CSP3	CSP7	GSP P16	
Positive	Negative	Positive	Negative	Positive	Negative	
Mucoepidermoid carcinoma	20	16	4	6	14	16	4	
Adenoid cystic carcinoma	1	1	0	1	0	1	0	
Salivary duct carcinoma	1	0	1	0	1	0	1	
Benign salivary tumor	12	2	10	1	11	1	11	

Figure 2 The representative negative results for FISH probes.

(A) FISH detection results of CSP3 (green) and GSP P16 (red) in normal cells. The white circle cell is enlarged in the inset, showing a normal signal pattern of 2 green and 2 red. (B) FISH detection results of CSP3 (green) and CSP7 (red) in normal cells. The white circle cell is enlarged in the inset, showing a normal signal pattern of 2 green and 2 red. The scale bar represents 10 μm.

Figure 3 The representative positive results for FISH probes.

(A) FISH detection results of CSP3 (green) and GSP P16 (red) probes in normal chromosome 3 and P16 gene deletion samples. The white circle cell is enlarged in the inset. The signal pattern shows 2 green and 0 red, indicating normal copies of chromosome 3 and homozygous deletion of the P16 gene. (B) FISH detection results of CSP3 and GSP P16 probes in polysomy chromosome 3 and P16 gene deletion samples. The white circle cell is enlarged in the inset. The signal pattern shows 3 green and 0 red, indicating chromosome 3 polysomy (three copies) and homozygous deletion of the P16 gene. (C) FISH detection results of CSP3 and CSP7 in cells with polysomies of chromosomes 3 and 7. The signal pattern of the green circle cell shows 4 green and 1 red, indicating chromosome 3 polysomy (four copies). The signal pattern of the red circle cell shows 2 green and 4 red, indicating chromosome 7 polysomy (four copies). (D) FISH detection results of CSP3 and GSP P16 probes in polysomy chromosome 3 and normal P16 gene cells. The white circle cell is enlarged in the inset. The signal pattern shows 4 green and 2 red, indicating chromosome 3 polysomy (four copies) and a normal P16 gene. The scale bar represents 10 μm.

Table 2 The sensitivity and specificity of individual and combined FISH probes for identifying benign and malignant salivary gland tumors in 34 tissue samples (22 malignant and 12 benign tumors).

Probe combination	Sensitivity	Specificity	
CSP3	77.3% (17/22)	88.1% (10/12)	
CSP7	31.8% (7/22)	91.6% (11/12)	
GSP P16	77.3% (17/22)	83.3% (10/12)	
CSP7+GSP P16	77.3% (17/22)	91.6% (11/12)	
CSP3+GSP P16	90.9% (20/22)	83.3% (10/12)	
CSP3+CSP7	77.3% (17/22)	75.0% (9/12)	
CSP3+CSP7+GSP P16	90.9% (20/22)	75.0% (9/12)	

The combination of CSP3 and GSP P16 probes in detecting malignant salivary gland tumors

The sample size was expanded to 78 to validate the sensitivity and specificity of CSP3, GSP P16, and their combination (Table 3). Among 38 malignant and 40 benign cases, CSP3 exhibited 23 true positives, 15 false positives, 36 true negatives, and four false negatives. Meanwhile, GSP P16 showed 28 true positives, 10 false positives, 34 true negatives, and six false negatives (Table S4). A sample was considered positive if the CSP3 or GSP P16 probe exhibited a fluorescent signal. The combined probes’ efficacy was assessed in the 78 clinical samples, with the results compared to actual pathological diagnoses. The receiver operating characteristic (ROC) analysis demonstrated sensitivities of 60.5% and 73.7%, specificities of 90.0% and 85.0%, and areas under the curves (AUCs) of 0.716 and 0.831 for CSP3 and GSP P16, respectively, in detecting malignant salivary gland tumors. When using the combined CSP3 and GSP P16 probes (with either probe positive), sensitivity increased to 89.5%, specificity remained at 85.0%, and the ROC area was 0.862 (Fig. 4).

Table 3 The sensitivity, specificity and AUC of individual and combined FISH probes for identifying benign and malignant salivary gland tumors in 78 tissue samples (38 malignant and 40 benign tumors).

Probe combination	Sensitivity	Specificity	AUC	
CSP3	60.5% (23/38)	90.0% (36/40)	0.716	
GSP P16	73.7% (28/38)	85.0% (34/40)	0.831	
CSP3+GSP P16	89.5% (34/38)	85.0% (34/40)	0.862	

Figure 4 ROC curves for the individual and combined detection using CSP3 and GSP P16 probes.

The results indicate that the P16 gene probe exhibits higher sensitivity, while the CSP3 probe shows superior specificity when used individually. Surprisingly, the combination of CSP3 and GSP P16 probes enhances sensitivity for malignant tumors, albeit with a slight reduction in specificity, thereby improving overall detection efficacy for malignant salivary gland tumors.

The clinical data of 78 patients (Table S5) were analyzed, among which 44.87% (35/78) had salivary gland tumors located in the parotid gland, followed by the submandibular gland at 17.95% (14/78) (Fig. S1A). In the group of 38 patients with malignant tumors, the parotid gland remained the most commonly affected, comprising 34.21% (13/38), followed by the respiratory tract and submandibular gland at 15.79% (6/38) (Fig. S1B). When further classifying tumor locations, the major salivary glands, consisting of the parotid, submandibular, and sublingual glands, were revealed to be more frequently involved, accounting for 66.67% of all salivary gland tumors and 55.26% of malignant salivary gland tumors (Figs. S1C–S1D). Moreover, no substantial difference was observed in the age distribution between patients with benign (49.38; 95% confidence interval (CI) [43.47–55.28]) and malignant salivary gland tumors (48.74; 95% CI [39.18–53.42]) (Fig. S1E).

Discussion

The incidence of salivary gland tumors has steadily increased in recent years. However, the diverse classification of these tumors and the morphological overlap between benign and malignant variants pose significant challenges in diagnosis. The lack of comprehensive, sufficient clinical data has hindered research and slowed progress in developing accurate diagnostic criteria to differentiate between benign and malignant forms. These diagnostic challenges complicate clinical decision-making and reduce prognostic accuracy for patients with salivary gland tumors. Therefore, there is an urgent need for more objective and standardized diagnostic methods to aid pathologists in their evaluations.

Polysomies and genomic imbalance are early hallmark features of tumorigenesis in cancer cells. Fluorescence in situ hybridization (FISH), developed to address these tumor characteristics, has been widely applied in the pathological diagnosis of urothelial tumors, particularly bladder cancer. However, the mechanisms and applications of FISH in salivary gland tumors remain underexplored. It provides objective, precise data for the pathological diagnosis of salivary gland tumors, minimizing the subjective variability frequently seen among pathologists. Our research analyzed 78 clinical samples, comparing the effectiveness of three probes, CSP3, CSP7, and GSP P16, in identifying salivary gland tumors in conjunction with clinical data. This study demonstrates that mutations in chromosomes 3 and 7, and alterations in the P16 gene, are crucial for the risk stratification of salivary gland tumors. Notably, CSP7 exhibited reduced sensitivity, while CSP3 and GSP P16 proved to be critical biomarkers for tumor malignancy. These findings establish a reliable basis for distinguishing benign and malignant salivary gland tumors, facilitating personalized treatment, risk assessment, and management strategies.

However, due to the absence of follow-up data, this study could not assess the impact of polysomies and genomic imbalances on prognostic factors such as postoperative recurrence in malignant salivary gland tumors. Future research will incorporate follow-up data to further explore the role of polysomies in tumor malignancy and prognosis, providing deeper insights into risk stratification.

Supplemental Information

Supplemental Information 1 Supplementary Materials.

Supplemental Information 2 Evaluation of FISH with CSP3, CSP7, and GSP P16 probes in 34 salivary gland tumor tissue samples.

The red mark represents positive cases.

Supplemental Information 3 Evaluation of FISH with CSP3 and GSP P16 probes for identifying benign and malignant salivary gland tumors in 86 tissue samples.

The red mark represents positive cases.

Supplemental Information 4 Clinical data of salivary gland tumor cases.

The collected clinical samples along with the corresponding clinical data including tumor histological type, location, patient age, and diagnosis time.

We thank Bullet Edits Limited for the linguistic editing and proofreading of the manuscript.

Additional Information and Declarations

Competing Interests

The authors declare that they have no competing interests.

Author Contributions

Yanxia Wu conceived and designed the experiments, performed the experiments, prepared figures and/or tables, and approved the final draft.

Zhiyue Xie conceived and designed the experiments, prepared figures and/or tables, and approved the final draft.

Nan Peng analyzed the data, authored or reviewed drafts of the article, and approved the final draft.

Rui Zhou analyzed the data, prepared figures and/or tables, and approved the final draft.

Liang Zhao conceived and designed the experiments, authored or reviewed drafts of the article, and approved the final draft.

Human Ethics

The following information was supplied relating to ethical approvals (i.e., approving body and any reference numbers):

All the experiments involved in this study comply with the Declaration of Helsinki, and approved by vote PJKT2024-196 of the Ethics Committee of Guangdong Medical University.

Data Availability

The following information was supplied regarding data availability:

The raw data is available in the Supplemental File.

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
