# Peer review of "Association of chromosomal aberrations in chromosomes 3 and 7, and P16 mutations with malignancy in salivary gland tumors"

_PeerJ, doi:10.7717/peerj.19217_

## Round 0.1 · original submission · Major Revisions

I kindly request that the valuable reviewers' suggestions be carefully considered.

Reviewer 1 ·

Basic reporting

The manuscript by Wu et al, entitled “The Association of Chromosomal Aberrations in Chromosomes 3 and 7, and P16 Gene Mutations with Malignancy in Salivary Gland Tumors”, explores the FISH-based approach in the diagnosis of salivary gland tumors, and demonstrates that the combination of Chr7 numerical abnormalities with the deletion of P16 serves as both highly specific and highly sensitive way to to discriminate between benign and malignant salivary gland tumors. While being technically, experimentally and conceptually sound, the study suffers a bit from the quality of the data presentation, which might be substantially improved (please see below).
The reviewer suggests to address following points:
1) Concepts of specificity and sensitivity in diagnostic testing (and a specific way to address both in the current study) need to be properly introduced, perhaps in the form of a separate Figure.
2) More information regarding the nature of numerical abnormalities along with the quantification criteria (going beyond what is included in lines 133-136!) needs to be presented (% of trisomies, monosomies, how the polysomies were defined etc). The thresholds (10%, 20%) need to be justified. Regarding the p16 deletion, did the authors consider homozygous deletion or the loss of one copy?
3) Figures need to be improved! General suggestions for all figures: the images are fairly dim, and need to be better contrasted. The authors should present enlarged insets (including one cell or a metaphase, depending on the panel) along low magnification images to make sure that the signals are easily captured by the reader. The figures should be better annotated, including names of probes (or chromosomes) presented in corresponding font colors. It will be also important to add scale bars. The figure legends should be expanded to include the information regarding the probe used and a corresponding color (for example, in Figure 1, ‘The FISH detection results of CSP3 (green) and GSP P16 (red) in normal cells’).
4) Figure 1: images shown in (A) and (C) are derived from the same sample, it should be stated in the legend (and showing the merged image will help the cause too, as the signals of different centromeric probes are located to different chromosomes!).
5) Figure 2: the images are of very low quality, heavily pixelated – please improve. Adding enlarged insets is crucial for understanding the figure.
6) Figure 3A: there are red dots (p16?) in the sample, which is stated to be P16-deleted, please clarify.
7) Table 2: please include the numbers of benign (12?) and malignant (22?) tumors in the title.
Minor points:
1) Please check the typos/English throughout the manuscript, i.e. lines 57, 59, 71 (perhaps ‘control of cell division’ might be used instead of ‘division controlling’) etc.
2) lines 153-155 and 167 are redundant (might be compressed into one sentence?).

Experimental design

please see above

Validity of the findings

please see above

Reviewer 2 ·

Basic reporting

1) The language level needs to be improved. Some passages of the text are difficult to understand.
2) The authors say to apply to salivary gland tumors observations obtained from other tumors, regarding chromosomal abnormalities 3 and 7 and p16 abnormalities. The authors should report in the introduction what is already known about these chromosomal aberrations in salivary gland tumors.

Experimental design

1) The sample size is not clear. The authors initially stated that FISH was performed on 34 tumors (line 146), but they applied FISH to 86 cases. the authors should better clarify the inclusion and exclusion criteria, and why they used two different series in these two phases of the research.
2) Authors should report false positives, true positives, true negatives, and false negatives, in the overall series, and in benign and negative tumors.
3) The authors should provide more details on the clinical and pathological characteristics of the series (histotypes of the neoplasms, location, age of the patient, etc.).
4) Are there any pediatric cases? The application of FISH on pediatric cases could be particularly interesting, due to the diagnostic difficulties in these patients for the diagnosis of low-grade mucoepidermoid carcinoma, which is also the most frequent salivary neoplasm in this age group (Ronchi A et al. Fine-Needle Aspiration Cytology Is an Effective Diagnostic Tool in Paediatric Patients with Mucoepidermoid Carcinoma as Secondary Neoplasm. Acta Cytol. 2020;64(6):520-531. doi:10.1159/000508395).

5) the authors should specify the year in which the cases were diagnosed, being a retrospective case study.

Validity of the findings

the discussion needs to be broadened and enriched.
1) The authors should state, in the discussion, data already known in the literature regarding chromosomal aberrations in salivary gland tumors, with a review of the pertinent literature.
2) the authors should explain what the main diagnostic difficulties of these tumors are, and what role the proposed FISH evaluation could have.

---

## Round 0.2 · Major Revisions

I would like to ask you to carefully consider the revisions requested by the reviewers in the article.

Reviewer 1 ·

Basic reporting

The authors have successfully addressed all of my comments/concerns, in particular, regarding the data presentation.

Experimental design

The authors have successfully addressed all of my comments/concerns, in particular, regarding the data presentation.

Validity of the findings

The authors have successfully addressed all of my comments/concerns, in particular, regarding the data presentation.

Reviewer 2 ·

Basic reporting

The language quality needs to be improved.
In my previous report, I suggested that the authors should report in the introduction what is already known about these chromosomal aberrations in salivary gland tumors (chromosome 3 and 7). The authors have already added a paragraph in the introduction about chromosomal aberrations in salivary gland tumors, other than chromosomes 3 and 7. Please, the authors should report in the introduction what is already known about these chromosomal aberrations in salivary gland tumors (chromosome 3 and 7, no other chromosomes!) (e.g. Götte K, Ganssmann S, Affolter A, et al. Dual FISH analysis of benign and malignant tumors of the salivary glands and paranasal sinuses. Oncol Rep. 2005;14(5):1103-1107.)

Experimental design

1. Thank you for clarifying the selection of cases in the rebuttal letter. However, upon reading the manuscript, it remains unclear that the probes were initially applied to a smaller subset of cases and then to the complete cohort. It is not clear what criteria were used to select the initial 34 cases. I have significant concerns regarding the histotypes, as there is mention of squamous carcinoma (which is very rare in salivary glands and more likely to be metastases) and even an ameloblastoma, which is not a neoplasm typically expected in salivary glands. It would be better to exclude these cases from the study.
2. The authors should clearly report the number of false positives and false negatives in the series.
3. The supplementary material includes many interesting databases. It would enhance the manuscript if the authors analyzed this data and incorporated it into the main text. Therefore, the authors should report the key clinical data of the series in the text (such as the most frequent sites, the age range of the subjects, etc.).

Validity of the findings

no comment

---

## Round 0.3 · accepted · Accept

I was pleased that the reviewers' requests for revision were fulfilled.

Reviewer 2 ·

Basic reporting

no comment

Experimental design

no comment

Validity of the findings

no comment

Additional comments

The authors have modified the text according to the previous comments. I have no further comments to make.